# Grape Lipidomics: An Extensive Profiling thorough UHPLC-MS/MS Method

**DOI:** 10.3390/metabo11120827

**Published:** 2021-11-30

**Authors:** Domenico Masuero, Domen Škrab, Giulia Chitarrini, Mar Garcia-Aloy, Pietro Franceschi, Paolo Sivilotti, Graziano Guella, Urska Vrhovsek

**Affiliations:** 1Research and Innovation Centre, Department of Food Quality and Nutrition, Fondazione Edmund Mach, 38098 San Michele all’Adige, Italy; domenico.masuero@fmach.it (D.M.); domen.skrab@gmail.com (D.Š.); mar.garcia@fmach.it (M.G.-A.); urska.vrhovsek@fmach.it (U.V.); 2Department of Agricultural, Food, Environmental and Animal Sciences, University of Udine, 33100 Udine, Italy; paolo.sivilotti@uniud.it; 3Unit of Computational Biology, Research and Innovation Centre, Fondazione Edmund Mach, 38098 San Michele all’Adige, Italy; pietro.franceschi@fmach.it; 4Bioorganic Chemistry Laboratory, Department of Physics, University of Trento, 38123 Trento, Italy; graziano.guella@unitn.it

**Keywords:** grape, lipidomics, liquid chromatography, mass spectrometry, lipidome

## Abstract

Lipids play many essential roles in living organisms, which accounts for the great diversity of these amphiphilic molecules within the individual lipid classes, while their composition depends on intrinsic and extrinsic factors. Recent developments in mass spectrometric methods have significantly contributed to the widespread application of the liquid chromatography-mass spectrometry (LC-MS) approach to the analysis of plant lipids. However, only a few investigators have studied the extensive composition of grape lipids. The present work describes the development of an ultrahigh performance liquid chromatography tandem mass spectrometry (UHPLC-MS/MS) method that includes 8098 MRM; the method has been validated using a reference sample of grapes at maturity with a successful analysis and semi-quantification of 412 compounds. The aforementioned method was subsequently applied also to the analysis of the lipid profile variation during the Ribolla Gialla cv. grape maturation process. The partial least squares (PLS) regression model fitted to our experimental data showed that a higher proportion of certain glycerophospholipids (i.e., glycerophosphoethanolamines, PE and glycerophosphoglycerols, PG) and of some hydrolysates from those groups (i.e., lyso-glycerophosphocholines, LPC and lyso-glycerophosphoethanolamines, LPE) can be positively associated with the increasing °Brix rate, while a negative association was found for ceramides (CER) and galactolipids digalactosyldiacylglycerols (DGDG). The validated method has proven to be robust and informative for profiling grape lipids, with the possibility of application to other studies and matrices.

## 1. Introduction

As stated by the Consortium of Lipid Metabolites and Pathways Strategy (Lipid MAPS), lipids can be defined as small amphiphilic or hydrophobic molecules that are insoluble or partially soluble in water [1]. Their role is essential for all eukaryotic and prokaryotic organisms, since they affect the cellular membrane structures and protein–membrane interactions, provide a source of energy through oxidation processes and serve as signaling molecules [2].

Ketoacyl and isoprene groups represent two fundamental building blocks for carbanion-based condensations of ketoacyl thioesters and/or by carbocation-based condensations of isoprene units, which classifies lipids into eight categories: fatty acyls (FA), glycerolipids (GL), glycerophospholipids (GP), sphingolipids (SP), saccharolipids (SL), polyketides (PK), sterols (ST), and prenol lipids (PR) [3]. Each category is further divided into classes and subclasses, which substantially leads to a wide-ranging diversity of lipid species in complex biological matrices, also known as lipidome [3,4].

Due to the chemical complexity and wide concentration range of the lipids present in biological matrices, to identify and potentially quantify all lipids simultaneously with a single analytical strategy is a challenging task [1]. Nonetheless, mass spectrometry (MS) coupled with soft electrospray ionization (ESI) has proven to be a powerful technique to perform such a challenging analytical task, due to its unique superiority in terms of specificity, sensitivity, dynamic range, and throughput [5,6,7,8]. Reducing matrix effects in biological samples as well as limiting the complexity of analytes at the moment of detection have contributed to a more frequent use of the liquid chromatography–mass spectrometry (LC-MS) approach, despite the fact that the direct infusion MS strategies were initially prevalent in lipidomic research, due to their relative simplicity of operation and fast analysis [9,10]. In the field of lipidomics, the good reproducibility and high resolving power of high-performance liquid chromatography (HPLC) enhance the separation of almost all the lipid molecular species [5]. However, the development of stationary phases and ultrahigh performance LC (UHPLC) provided much higher resolution and made the process less time-consuming than traditional HPLC [11].

In recent years, MS-based lipidomics has proven to be a suitable approach for investigating the lipidome size in biological systems, where the entire collection of chemically distinct lipid species is included [12]. Since in this paper we are focusing on grape lipidomics, it is important to point out that the majority of plant lipids are similar to those present in mammals, except for some classes of galactolipids, sulfolipids, galactosyl group-containing sphingolipids, and plant sterols, which are only minimally present or even absent in mammals [13]. For example, diacylglycerolipids, such as monogalactosyldiacylglycerol (MGDG), digalactosyldiacylglycerol (DGDG) and sulfoquinovosyldiacylglycerol (SQDG), are involved in photosynthesis and are therefore most prominent in chloroplast membranes [14,15].

The vast majority of lipid molecules can be found in the membranes of plant cells and together they represent 5–10% of the dry weight of vegetative plant cells [16]. In grapes, this quantity slightly increases, as the proportion of lipids falls within the range of 0.15–0.24% of the fresh weight [17], which makes them an important modulator of yeast metabolism, during the white and rosé winemaking process, due to the short contact of grape skins with must [16]. This ultimately affects the aromatic profile of the produced wines. For instance, it has been observed that a higher lipid concentration in the fermenting medium results in a lower production of acetic acid, whereas it stimulates 3-mercaptohexan-1-ol thiol liberation, which contributes to the increase of the perception of citrus notes in wine [18]. Similarly, polyunsaturated fatty acids (i.e., linoleic, and linolenic acid) may be oxidized by lipoxygenase-hydroperoxidelyase, thus forming C_6_ and C_9_ aldehydes (e.g., *trans*-2-hexenal, *trans*-2-nonenal, and *cis*-2-hexenol) that are related to green and herbaceous odors [19,20]. In order to promote yeast cell growth under hypoxic conditions, a sufficient degree of unsaturation in the yeast plasma membrane is required, since the biosynthesis of fatty acids is repressed as ethanol concentration in the environment increases [21,22,23]. The unsaturated fatty acids are therefore required for physiological functions related to yeast adaptation. The availability of these lipids, however, is related to the introduction of double bonds in the fatty acyl chains, catalyzed by desaturases [24]. Complex forms of exogenous GP and GL from grapes thus represent a potential source of free fatty acids (FA) for yeast utilization, which are liberated through lipolytic activity [25].

It can be therefore concluded that lipid molecules have an important physiological role in yeast cells, and consequently on the wine aroma. This can be a sufficient reason to explore the lipidome of grapes. In recent years, the investigators have focused on the characterization of predominantly fatty acids that are present in *Vitis vinifera* [17] and non-*V*. *vinifera* [26] grape cultivars. In addition, the evolution of fatty acids [27], phytosterols [28], and TG [29,30] in grapes during berry development has been established. However, with a few exceptions [16,31,32], relatively few studies addressed a more holistic lipid profiling of grapes, in order to better understand lipid metabolism.

The aim of this work is to introduce and validate a sensitive and accurate UHPLC-MS/MS method for the simultaneous determination and semi-quantification of multiple classes of lipids in grapes of the Ribolla Gialla variety. In addition, the effectiveness of this method is verified by monitoring the changes occurring in the lipid profile during grape ripening.

## 2. Results and Discussion

### 2.1. Compounds of Interest

Since the method was designed to be validated and applied to a vegetal matrix, it was decided to consider only chemical backbones made up of an even number of carbons, from 14 to 22, and with up to six double bonds that configure the fatty acids as the building block of other lipid species [33]. Appendix A summarizes the total number of various saturated and unsaturated chains that have been detected in our reference sample. As can be observed, lipids with 18:2 and 16:0 fatty acid chains prevailed in our grape samples, which was also confirmed by the area percentage of chains found in the reference grape extract (Appendix A). As it has been previously reported in the literature [34], the plant tissues most often contain between 14 and 24 carbon atoms, which could confirm our decision regarding the chain length for the validation.

The instrument parameters were optimized using the standard mix as described in the Materials and Methods section; following LIPID MAPS^®^ recommendations [35], four categories of compounds were considered in the method, with their relevant Multiple Reaction Monitoring (MRM) transitions and parameters (Appendix A). For MRM selection, we followed the criteria described in Section 3.2.

We studied the fragmentation pattern for each class of compounds, taking into account the existing literature. Q1 scan spectra of each lipid showed different ion species, due to the presence or the absence of adducts (Table 1).

In detail, glycerophosphate (PA), glycerophosphoethanolamine (PE), glycerophosphoglycerol (PG), glycerophosphoinositol (PI), and glycerophosphoserine (PS) are identified in negative mode using [M–H]^−^ as precursor ion; instead, glycerophosphocholines (PC) are characterized by the adduct ion [M+HCOO]^−^ as the base peak; for all the glycerophospholipids the [sn2 FA]^−^ chain has been reported as product ion [33,36,37,38,39,40,41,42]. Lyso-glycerophosphate (LPA), lyso-glycerophosphoethanolamine (LPE), lyso-glycerophosphoglycerol (LPG), lyso-glycerophosphoinositol (LPI) and lyso-glycerophosphoserine (LPS) are characterized by a negative ionization using a [M–H]^−^ as a precursor with a [sn2 FA]^−^ fragment as a product ion [33,36,37,40,41,42]; instead, lyso-glycerophosphocholines (LPC) are characterized by a positive ionization [M+H]^+^ with a product ion characteristic of *m*/*z* 184.1 which represents the head group of phosphoryl choline [33,43]. Monoacylglycerols (MG) are characterized by a positive ionization mode [M+H]^+^ with a product that consists of the molecule after the loss of glycerol [M–C_3_H_7_O_3_]^+^ [33,44]. In addition, diacylglycerol (DG) and TG ionize in positive mode with a sodium adduct [M+Na]^+^, the product ion being the molecule after the loss of [M–(sn2 FA)]^+^ for DG and [M–(sn3 FA)]^+^ for TG [33,44,45]. MGDG and DGDG are identified with a positive ionization mode with the sodium adduct [M+Na]^+^ and a loss of the fatty acid with a rearrangement of the molecule for the product ion [M+Na–R_2_CO_2_H]^+^, as described by Guella et al. [46]. Sphingomyelins (SM) have been included with a positive ionization [M+H]^+^ as precursor ion and a *m*/*z* 184.1 as product ion [33,47]. CER are ionized in a positive mode with the abduction of a molecule of water [M+H–18]^+^ with a product ion of *m*/*z* 264.1 for CER, glucosyl ceramide (glcCER) and lactosyl ceramide (lacCER) and of *m*/*z* 266.1 for all the dihydroceramide (dhCER) [33,48]. Carnitine (CAR) ionizes in a positive mode [M+H]^+^ with a product fragment of *m*/*z* 85.1, which was described as a specific product ion of acylcarnitine fragmentation [33,49]. FA ionization polarity is in negative mode with [M–H]^−^ as Q1 and as the entire molecule without fragmentation as product ion [33,50].

Among the 8098 MRM considered (included 21 internal standards, IS), 1045 were detected in our reference matrix (grapes at the final maturation point). Due to the extensive number of MRM, two independent analytical methods for grapes samples were built: one in positive and one in negative ionization mode. In this way, we avoided polarity switch, guaranteeing a suitable number of points per peak along the chromatographic run.

**Table 1 metabolites-11-00827-t001:** Fragmentation pattern for each class of lipid compounds.

Class	Ionization Mode	Precursor Ion	Product Ion	Reference	Internal Standard	DP (Volts)	EP (Volts)	CE (Volts)	CXP (Volts)
CAR	pos	[M+H]^+^	85.1	[33,49]	24:0 (d4) Carnitine	93	10	31	16
CER	pos	[M+H–18]^+^	264.1	[33,48]	C15 Ceramide-d7	130	10	55	10
DG	pos	[M+Na]^+^	[M– (*sn*2 FA)]^+^	[33,44]	15:0–18:1(d7) DG-Na	93	9	42	25
DGDG	pos	[M+Na]^+^	[M+Na–R_2_CO_2_H]^+^	[46]	Hydrog DGDG (18:0–18:0)	80	10	65	20
dhCER	pos	[M+H–18]^+^	266.1	[33,48]	C15 Ceramide-d7	130	10	55	10
FA	neg	[M–H]^−^	[M–H]^−^	[33,50]	Stearic acid-d3	−80	−10	−17	−20
glc-dhCER	pos	[M+H–18]^+^	266.1	[33]	C15 Ceramide-d7	130	8	45	27
glcCER	pos	[M+H–18]^+^	264.1	[33]	C15 Ceramide-d7	130	8	45	27
lac-dhCER	pos	[M+H–18]^+^	266.1	[33]	C15 Ceramide-d7	126	10	56	15
lacCER	pos	[M+H–18]^+^	264.1	[33]	C15 Ceramide-d7	126	10	56	15
LPA	neg	[M–H]^−^	[*sn*2 FA]^−^	[33,41]	17:0 Lyso PA	−80	−6	−45	−20
LPC	pos	[M+H]^+^	184.1	[33,43]	18:1(d7) Lyso PC	90	6	35	20
LPE	neg	[M–H]^−^	[*sn*2 FA]^−^	[33,40]	18:1(d7) Lyso PE	−88	−12	−42	−20
LPG	neg	[M–H]^−^	[*sn*2 FA]^−^	[33,42]	17:1 Lyso PG	−75	−10	−38	−24
LPI	neg	[M–H]^−^	[*sn*2 FA]^−^	[33,36]	17:1 Lyso PI	−90	−6	−40	−24
LPS	neg	[M–H]^−^	[*sn*2 FA]^−^	[33,37]	17:1 Lyso PS	−72	−10	−53	−24
MG	pos	[M+H]^+^	[M–C_3_H_7_O_3_]^+^	[33,44]	18:1(d7) MG	140	10	16	10
MGDG	pos	[M+Na]^+^	[M+Na–R_2_CO_2_H]^+^	[46]	Hydrog MGDG (18:0–16:0)	100	10	50	30
PA	neg	[M–H]^−^	[*sn*2 FA]^−^	[33,40,41]	15:0–18:1-D7-PA	−80	−6	−45	−20
PC	neg	[M+HCOO]^−^	[*sn*2 FA]^−^	[33,38,39,40]	15:0–18:1(d7) PC	−90	−10	−50	−20
PE	neg	[M–H]^−^	[*sn*2 FA]^−^	[33,38,40]	15:0–18:1(d7) PE	−88	−12	−42	−20
PG	neg	[M–H]^−^	[*sn*2 FA]^−^	[33,40,42]	15:0–18:1(d7) PG	−75	−10	−38	−24
PI	neg	[M–H]^−^	[*sn*2 FA]^−^	[33,36,40]	15:0–18:1(d7) PI	−50	−10	−55	−10
PS	neg	[M–H]^−^	[*sn*2 FA]^−^	[33,37,40]	15:0–18:1(d7) PS	−72	−10	−53	−24
SM	pos	[M+H]^+^	184.1	[33,47]	d18:1–18:1(d9) SM	124	10	32.5	23
TG	pos	[M+Na]^+^	[M– (*sn*3 FA)]^+^	[33,44,45]	15:0–18:1(d7)-15:0 TG-Na	90	10	40	10

Declustering potential, DP; entrance potential, EP; collision energy, CE; collision cell exit potential, CXP.

### 2.2. Chromatographic Optimization

In order to optimize the method that includes such a large number of compounds, the chromatographic separation was carefully tested with three different columns and two chromatographic methods. To evaluate the quality of the chromatographic separation, positive and negative mode graphs with *m*/*z* and retention time were built (Figure 1). The graphs show how the compound classes are distributed along the separation time according to their *m*/*z* for the three types of columns. The XBridge Amide column, which was recently used by Khan et al. [51] for the lipidomic analyses of human plasma, did not guarantee, in our application, a suitable separation of TG, DG and MGDG, which represent around 60% of the compounds identified in our matrix (Figure 1).

The AquityBEH-C18 and AquityCSH-C18 columns, used following Isaac et al. [52], showed a better separation potential. In terms of numbers, both columns gave us a good separation of those critical compounds (TG, DG and MGDG), which appear spread along the chromatographic time (Figure 1). Between the two we finally chose the Aquity CSH-C18 column, due to the best peak shape obtained (data not shown).

### 2.3. Method Validation

The compound annotation has been evaluated using the Kendrick mass defect (KMD) to the CH_2_ base [53]. The KMD value of each compound has been plotted versus the specific retention time (Appendix A). Following this validation method, compounds with the same number of carbons should be aligned in the same diagonal and compounds with the same number of double bounds should be horizontally aligned. We decided then to exclude the compounds that do not reflect these rules (falling out of the diagonal and falling out of the horizontal line). The method was validated using either the IS mix or the reference matrix sample as described in Section 3.5. Table 2 shows the number of compounds per class included in the method, the validation parameters evaluated using the IS mix, the number of compounds identified in our reference matrix and those validated.

The final value of recovery reported in Table 2 was calculated as the mean of the three recovery level points for each class of compounds (% within a single class). As shown in the table, the recovery was over 80% for 16 classes (CAR, CER, DG, DGDG, FA LPA, LPC, LPE, MG, MGDG, PA, PC, PE, PG, PS, SM and TG), 76% for LPA, and 68% for PI; we obtained a good recovery for almost all the classes except for LPG (29%), LPI (4%) and LPS (39%). The Folch extraction method consists of a mixture of chloroform, methanol, and water, therefore some hydrophilic compounds can be contained in the aqueous phase [54]. Regardless, we obtained good recovery results for the majority of classes of our compounds of interest. We are aware that the extraction method used was a sort of compromise and cannot be optimized for all the compound classes together, but this situation has to be accepted to ensure a large coverage of the different lipid classes.

The linearity range for each class of compounds was evaluated using the IS mix calibration curves spiked to the matrix; the ranges are reported in Table 2. The limit of quantification (LOQ) was identified as the lowest point of the calibration curve included in the linearity range; the limit of detection (LOD) was calculated as 3.3 × (S/N).

From the point of view of repeatability (six independent extraction) of the 1045 compounds detected in the reference matrix, we decided to include only those 412 with a repeatability value of CV ≤ 20%, following [55], taking into account the KMD validation. We evaluated the intra-day and inter-day repeatability of those using one extract of the reference matrix injected six times consecutively during the same day and performing three injections per day during seven consecutive days. CV% should not exceed a value of 15% for intra-day assay and 20% for inter-day assay [55]. We are aware that LPG, LPI and LPS had shown low recovery values due to the extraction method used in this application, which is not the most suitable for these classes of compounds; however, their repeatability and stability gave good overall results. For this reason, we believe they can be used for the comparison of our samples, and we therefore decided to consider them.

Appendix A shows the intra-day and inter-day repeatability values for the successfully validated compounds.

### 2.4. Method Application for Grape Maturation Samples

Appendix A provides the semi-quantification of the 412 identified and validated lipid compounds in the samples, expressed as µg/g of fresh grape powder. In this benchmark experiment, the proposed lipidomics method was applied to characterize the ripening process of Ribolla Gialla grapes. An exploratory analysis of the lipidomic dataset was performed through Principal Component Analysis (PCA, Figure 2). The scores plot (Figure 2A) shows that the first two principal components accounted for 54% of the total variance. In the plot, it can be observed that the quality control (QC) samples (indicated in grey) are clustered together, close to the origin of the scores plot. This indicates that their variability was lower than that observed among the study samples. This is a clear indication of a good analytical reproducibility. In terms of study samples, a certain degree of separation between the samples corresponding to the first (from maturation point 1 to 7) and second half of the maturation period (from maturation point 8 to 13) was observed. In particular, the points that tend to cluster in the upper left side of the plot correspond to the initial maturation samples (the maturation time-point is indicated by the numbers under each sample from 1 to 13), whereas the samples corresponding to the last ripening stages extend mainly across the bottom right part. The global picture is less clear in the case of the loadings plot (Figure 2B). There, it is difficult to find any specific trend that could directly connect certain classes of lipids to an observed ripening trend. The only exception appears to be that the TG seem to be located on the right side of PC1, where the samples from the second half of the maturation process tended to concentrate.

In order to pinpoint the lipids which were strongly associated with the ripening process, a more powerful regression approach was applied. The idea was to use Partial Least Squares (PLS) regression to find the lipids which can be used to predict the °Brix value (a measure associated with the fruit maturation state). Model optimization resulted in a two-component model able to explain a substantial amount of °Brix variability (R^2^ = 0.889, median Q^2^ = 0.855).

To get a general overview of which types of compounds were more strongly associated with the °Brix, the regression coefficients of the validated PLS model were taken into consideration. The regression coefficient of each lipid in the model is, indeed, giving a measure of its association with the °Brix level. Large positive values speak of positive association, while large negative values are the indication of negative association.

To work at the lipid class level, one could check if the lipids belonging to a specific class were prevalently positively or negatively associated with the maturation, by checking if the majority of them was getting a relatively large positive (or negative) regression coefficient. In order to do that, we used the first and the last quartile of the distribution of the regression coefficients as thresholds to define large positive or negative association. The results of this type of analysis are summarized in Figure 3. Figure 3A displays the distribution of the regression coefficients and the thresholds for the first and fourth quartiles. Figure 3B, instead, shows the fraction of lipids of each specific class which were getting regression coefficients in the upper or lower quartile. In the case of CER, for example, around 60 percent of the lipids were showing a regression coefficient in Q1 (negative association). This is an indication that ceramides are “globally” negatively associated with maturation. This resulting regression lines for a subset of the more interesting lipid classes are presented in Figure 3C.

In general, we considered that the classes of lipids showing a higher proportion of individual compounds in the first quartile (Q1, i.e., those with the lowest regression coefficients) would represent the categories with a general decreasing behavior over the maturation process, whereas those classes with higher proportions of individual compounds in the fourth quartile (Q4) would indicate the lipid classes with a direct connection to the sugar level of the grapes (i.e., maturation process).

Figure 3 shows that the classes of CER, MG, LPG, DGDG and MGDG were negatively related to the increasing °Brix value, whereas LPC, LPE, PE, PG and PI were globally growing with the °Brix value. For example, 55% (*n* = 20) of MGDG were allocated in Q1, while 16% (*n* = 6) had a regression coefficient in Q4, and 20% (*n* = 7) and 50% (*n* = 17) of PE were distributed in Q1 and Q4, respectively (Appendix A).

The fluctuation of “neutral lipids” (i.e., TG, DG, and FA) has been confirmed with some previous studies. Namely, Barron et al. [30] found out that TG with unsaturated fatty acid residues changed significantly towards the end of the grape ripening process, but without a clear trend, which could explain our findings regarding TG positioning in the second or third quartile groups. Plants synthesize fatty acids and store them as TG in seeds, and subsequently utilize them as energy during seed germination and early seedling development [56,57]. However, stressful conditions, such as cold, heat, mechanical wounding, and phosphorus deficiency [13] can cause the activation of lipases that substantially liberate fatty acids from molecules and convert them to acetyl-CoA units, which are precursors of sugar synthesis. [56]. Given the fact that TG is the most abundant class of lipids also in grape seeds [58], it can therefore only be speculated that the fluctuations of the neutral lipids are caused by stress factors.

Among the compounds whose concentration increased with grape maturity, we identified phospholipids PG and PE (Figure 3B) which are common constituents of biological membranes. Between the two, PG is of particular importance, since it makes up the lipid bilayer of the thylakoid membrane in plant chloroplasts, together with galactolipids (i.e., DGDG and MGDG) and sulfolipids (i.e., SQDG) [59].

CER and DGDG turned out to be two major classes of lipids showing a general decrease during grape maturation. The wide range of sphingolipid structures enables their function in a variety of cellular processes: from acting as structural integrity elements of the membrane to mediating cellular processes such as programmed cell death, which can be promoted by CER accumulation [60,61]. Moreover, the decreasing trend of DGDG could be linked to the decreasing concentration of linolenic acid during grape maturation [62], indicating an important role of the lipoxygenase pathway in combination with galactolipases and phospholipases during the ripening period [63].

## 3. Materials and Methods

### 3.1. Chemicals

EquiSPLASH™ LIPIDOMIX^®^ Quantitative Mass Spec Internal Standard mix containing 13 deuterated compounds was purchased from Avanti^®^ Polar Lipids (Alabaster, AL, USA). All of the following were purchased from Avanti^®^ Polar Lipids (Alabaster, AL, USA): 17:0 lyso PA, 15:0–18:1-D7-PA, Hydrogenated MGDG (18:0–16:0), Hydrogenated DGDG (18:0–18:0), 17:1 lyso PG, 17:1 lyso PS, 17:1 lyso PI, 24:0 (d4) carnitine and stearic acid-d3.

The following mix and single compounds were purchased from Avanti^®^ Polar Lipids (Alabaster, AL, USA): Soy PA, Soy PC, Soy PE, Soy PG, Soy PI, Soy PS, Soy Lyso PI, Soy Lyso PC, MGDG, DGDG, Brain SM, Ceramide (Egg), C12 Carnitine, C16 Dihydroceramide (d18:0/16:0), C18 Glucosyl(ß) Ceramide (d18:1/18:0) and C24 Lactosyl(ß) Ceramide (d18:1/24:0); Lipid Standard Mono-, Di-, and Triglyceride Mix was purchased from Sigma-Aldrich (Milan, MI, Italy).

The chemicals acetonitrile (ACN, LC-MS grade), 2-propanol (IPA), methanol (CH_3_OH, LC-MS grade) and chloroform (CHCl_3_) were purchased from Sigma-Aldrich (Milan, MI, Italy). Formic acid (HCOOH) and ammonium formate (NH_4_COOH) additives for LC-MS were from FLUKA Sigma-Aldrich (Milan, MI, Italy). Purified water was used for the extraction procedure and mobile phase preparation using a Milli-Q system (Millipore, Milan, Italy). During the extraction procedure butylated hydroxytoluene (BHT) was used as antioxidant, provided by Aldrich-Fluka-Sigma S.r.l. (Milan, MI, Italy).

### 3.2. Compounds of Interest and Their Characteristics

Four categories of compounds were considered for this method: glycerophospholipids: PA, PC, PE, PG, PI, and PS; LPA, LPC, LPE, LPG, LPI and LPS; glycerolipids: MG, DG, and TG, MGDG and DGDG; sphingolipids: SM, CER, glcCE, lacCER, dhCER and fatty acids: CAR and FA.

The MRM transitions were built by studying the chemical characteristics of each class of compounds, using the corresponding class compounds included in the standard mix. MRM covers all the possible combinations in a range of even carbon numbers from 14 to 22, with the double bond possibilities ranging from 0 to 6. Following the criteria described above, we obtained 35 combinations for the single-chain compounds and 630 combinations for those compounds having two chains; for the three-chain compounds, 7700 combinations are possible; however, the number was reduced to 1834 Q1/Q2 unique combinations reported in Appendix A. As reported by Michaelson et al. [60], the SM and its precursor CER are mainly characterized by an 18:1 fatty acyl chain; following this consideration we evaluated them as single-chain compounds (with a constant 18:1 fatty acyl chain) with the inclusion of 35 possible Q1/Q2 combinations (Appendix A).

A detailed list of MRMs is shown in Appendix A, including compound class, ionization mode, precursor ion, product ion and the mass spectrometry parameters (declustering potential, DP; entrance potential, EP; collision energy, CE; and collision cell exit potential, CXP).

### 3.3. Instrumental Conditions

#### 3.3.1. Optimization of Liquid Chromatography Conditions

Due to the large number of compounds of interest and their chemical diversity, LC conditions were optimized.

The first tested method was set up with an XBridge Amide HPLC column (4.6 × 150 mm, 3.5 μm) (Waters, Milford, MA, USA) [51].

Separation was carried out with an initial flow of 0.7 mL/min following the gradient: 0 min 0.1% B; 0–6 min 6% B; 6–10 min increase to 25% B; 10–11 min 98% B; 11–13 min 100% B, held until 18.6 min. Then at 18.6–18.7 min the percentage of B decreased to 0.1% and maintained until 24 min. At 13.5 min the flow changed to 1.5 mL/min; then the flow returned to 0.7 mL/min at 23.5 min. The column was kept at 35 °C; the total duration of the analysis was 24 min. The mobile phase A consisted of 1 mM ammonium acetate in 5:95 water/acetonitrile (*v*/*v*) solution with pH 8.4, while the mobile phase B consisted of 1 mM ammonium acetate in 50:50 water/acetonitrile (*v*/*v*) solution, with pH 8.2.

For the second tested method, the Waters’ application note was considered, with some modifications [52]. The column was an Acquity UPLC CSH-C18 (2.1 × 100 mm, 1.7 µm) (Waters, Milford, MA, USA) at a flow rate of 0.6 mL/min. The mobile phase A consisted of 60:40 (*v*/*v*) H_2_O:ACN, 10 mM NH_4_COOH, and 0.1% HCOOH; B 90:10 (*v*/*v*) IPA:ACN 10 mM NH_4_COOH, and 0.1% HCOOH. The separation gradient was: 0–0.5 min 10% B; 0.5–22 min 97% B maintained for 4 min; 26–26.5 min 10% B held for 3.5 min. The Acquity UPLC BEH-C18 column (2.1 × 100 mm, 1.7 µm) (Waters, Milford, MA, USA) was also tested by applying this chromatographic method, as suggested by Cajka and Fiehn [9]. The column was kept at 55 °C; the total duration of the analysis was 30 min. The injection volume for all tested methods was 5 µL.

#### 3.3.2. Mass Spectrometry Parameters

The AB SCIEX Triple Quad™ 6500 LC-MS/MS system was used in positive and negative ESI mode. The mass analyzer conditions were optimized for each class of compounds.

The ion spray voltage was set at 5500 V for positive mode and −4500 V for negative mode. The source temperature was set at 500 °C, the nebulizer gas (Gas 1) and heater gas (Gas 2) at 50 and 60 psi, respectively. Instrument control and data acquisition were performed by Analyst software version 1.7 (Applera Corporation, Norwalk, CT, USA).

### 3.4. Sample Collection and Lipid Extraction

Ribolla Gialla (*V*. *vinifera*) grape samples originated from the Corno di Rosazzo vineyard site, located in the Friuli Colli Orientali and Ramandolo districts in the Friuli Venezia Giulia region of Northeastern Italy (46°00′19.1″ North; 13°26′30.6″ East; elevation 94 m a.s.l.). The clone used for the present study was the VCR 100 (Vivai Cooperativi Rauscedo, Rauscedo, PN, Italy) grafted onto the Kober 5BB rootstock. The training system adopted is a single arched Guyot. In order to determine how the lipid profile changes during grape ripening, the bunches were sampled regularly every 3–5 days from the véraison stage onwards. A total of 13 sampling points were collected from 6 August to 24 September 2019, each of which represented a single sample. Moreover, for each sampling point, five biological replicates were collected, with the exception of the last two points, where three replicas were available. Total soluble solids (°Brix) were measured using a manual refractometer (ATC-1, Atago, Tokyo, Japan).

Once the grape samples were collected and transferred to the laboratory, the berries with pedicels were randomly separated from different parts of grape bunch and frozen at −80 °C. A certain quantity of frozen grape berries without pedicels was subsequently homogenized under liquid nitrogen with an IKA A11 (Staufen, BW, Germany) homogenizer to generate ~30 g of powder, as previously described by Gika et al. [64]. Based on the Folch method [65], a pre-existing protocol [16] was followed for the extraction of the lipids from the grape powder, with some minor modifications. Briefly, 300 µL of CH_3_OH was added to the previously weighted 100 mg of frozen grape powder into an Eppendorf microtube, and the mixture was vortexed for 30 s. Afterwards, 600 µL of CHCl_3_, containing butylated hydroxytoluene (BHT 500 mg/L) was added, with 15 µL of IS (10 mg/mL), and placed in an orbital shaker for 60 min. Subsequently, 250 µL of Milli-Q purified H_2_O was added to the extracting mixture, followed by centrifugation at 3600 rpm for 10 min at 4 °C. The total lipid-rich layer was collected into the fresh Eppendorf microtube. For the second round of extraction, 400 µL of CHCl_3_/CH_3_OH/H_2_O 86:13:1 (*v*/*v*/*v*) was used, followed by centrifugation, extraction, and the addition of the lower lipid phase to the previously obtained lipid fraction. The remaining solvent present in the samples was then evaporated under a stream of N_2_, and the dried extracts were reconstituted in 300 μL of IPA. A QC sample was created by pooling the samples using 20 µL of each extract, used as QC and injected in the same sequence as the samples.

### 3.5. Method Validation

To ensure a confident lipid identification, we manually curated lipid annotations by plotting the retention time of a given lipid species against its KMD value to the CH_2_ base [53]. The IS mix was used to assess the recovery and to build the calibration curves determining the range of linearity and limit of detection (LOD).

For repeatability and stability, a reference matrix sample was used, corresponding to the grounded mature grape of Ribolla Gialla (final point of maturation stage, see Section 3.4). The method was validated according to the currently accepted US Food and Drug Administration (FDA) bio-analytical method validation guide [55].

#### 3.5.1. Recovery

Recovery was estimated on reference matrix samples spiked using the IS mix at three different levels (0.1, 0.5 and 1 mg/L). Five replicates for each point were performed and the values were calculated as the average of the “measured value/expected value” ratio (%). The final value of recovery reported in Table 2 is represented by the mean of the three levels for each class of compounds (% within a single class).

#### 3.5.2. Linearity, Limit of Detection and Limit of Quantification

Calibration curves were made in the reference matrix extract by adding increasing concentrations of the IS mix in different concentrations. The calibration curves were used to evaluate the range of linearity for each class of compounds and their LOD and LOQ were expressed in µg/g of fresh grape.

#### 3.5.3. Repeatability

Repeatability was evaluated using six independent extractions of the reference matrix and expressed as coefficient of variation (CV%). The repeatability was evaluated on the compounds detectable in our reference matrix.

#### 3.5.4. Intra- and Inter-Day

Intra- and inter-day variability were evaluated using six extractions using the reference matrix merged in a single vial. To evaluate the intra-day variability 6 consecutive injections were performed during the same day. To evaluate the inter-day variability three injections per day were performed during 7 consecutive days. Intra-day and inter-day variability were evaluated by calculating the coefficients of variation (CV%).

### 3.6. Data Analysis

Data processing was performed using MultiQuant, version 3.0 (Sciex, Concord, Vaughan, ON, Canada). The compound semi-quantification was calculated using one internal standard for each class (Table 1) and normalized for the exact weight of grape powder.

Missing values were replaced by a random value between 0 and half of the corresponding minimum value for each compound. PCA was performed using the “prcomp” function of R (version 4.0.3) on log-transformed and Pareto-scaled data.

The PLS regression model was fitted using the “pls” R package on log-transformed, scaled, and centered data. The contribution of each variable (i.e., lipid compound) to the prediction of the °Brix in the PLS model was evaluated using the regressions coefficients. The number of components of the model was estimated by 10-fold cross-validation. The coefficient of multiple determination (R^2^) from the fitted model was used as a measure of prediction accuracy of the model. The predictive performance of the model on an independent test set was assessed by repeatedly (1000 times) splitting the data in training and test segments (2/3 training, 1/3 test). The median of the resulting Q^2^ was used as a global measure of model fit. As a further validation strategy, a label permutation approach was implemented. This last strategy was also applied on 1000 models assigning the °Brix degree randomly. This strategy resulted in a median Q^2^ of −0.033. The full set Q^2^ is plotted in Appendix A.

Heatmaps were plotted using the “pheatmap” R package.

## 4. Conclusions

Despite the reported difficulties in analyzing such a chemically complex class of compounds, in the present work we developed a UHPLC-MS/MS method that includes 8098 MRMs (including 21 IS). By exploring the liquid chromatographic separation and the available columns, we identified a good resolutive method to achieve the best results for such a large number of compounds supported by the MRM mass spectrometry. A total of 1045 compounds were detected in our reference matrix; 412 of these were successfully validated and semi-quantified in a grape ripening study involving the Ribolla Gialla variety. The presented method allows the semi-quantification of single compounds; our first purpose was to provide a general overview of the application results, presenting the 412 compounds by class. Different classes of compounds were identified as directly or inversely related to the Ribolla Gialla grape ripening, showing the success of the analytical platform. The method can be applied to different studies and matrices starting from the 8098 MRMs, following the detection, validation and semi-quantification steps as presented in this work and allowing a broad overview of the major lipid classes.

## Figures and Tables

**Figure 1 metabolites-11-00827-f001:**
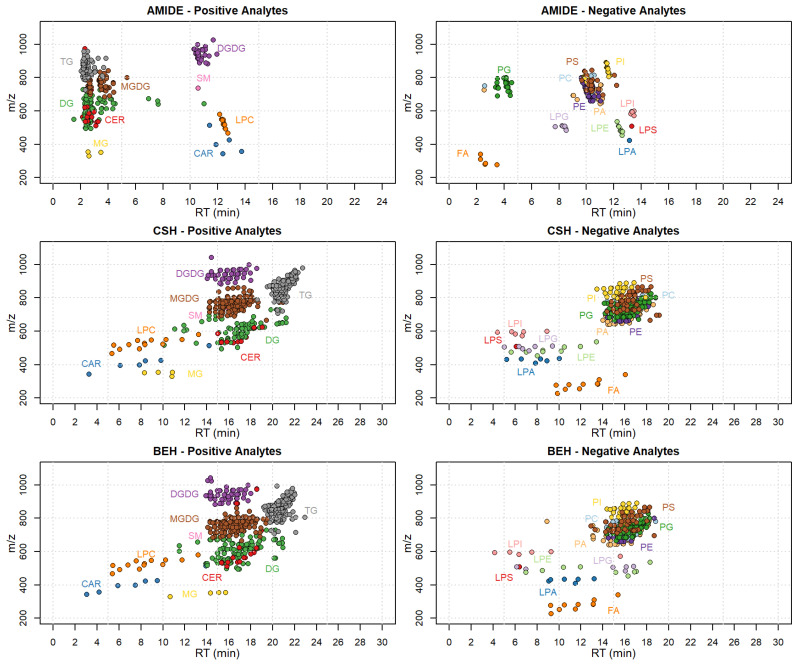
Distribution of the different compound classes along the retention time (RT) according to their *m*/*z*; the three tested columns (XBridge Amide column, Aquity CSH-C18 column and Aquity BEH-C18 column) are shown. Positive analytes: CAR, carnitines; CER, ceramides; DG, diacylglycerols; DGDG, digalactosyldiacylglycerols; LPC, lyso-glycerophosphocholines; MG, monoacylglycerols; MGDG, monogalactosyldiacylglycerols; SM, sphingomyelins; TG, triacylglycerols. Negative analytes: FA, free fatty acids; LPA, lyso-glycerophosphates; LPE, lyso-glycerophosphoethanolamines; LPI, lyso-glycerophosphoinositols; LPG, lyso-glycerophosphoglycerols; LPS, lyso-glycerophosphoserines; PA, glycerophosphates; PC, glycerophosphocholines; PE, glycerophosphoethanolamines; PI, glycerophosphoinositols; PG, glycerophosphoglycerols; PS, glycerophosphoserines.

**Figure 2 metabolites-11-00827-f002:**
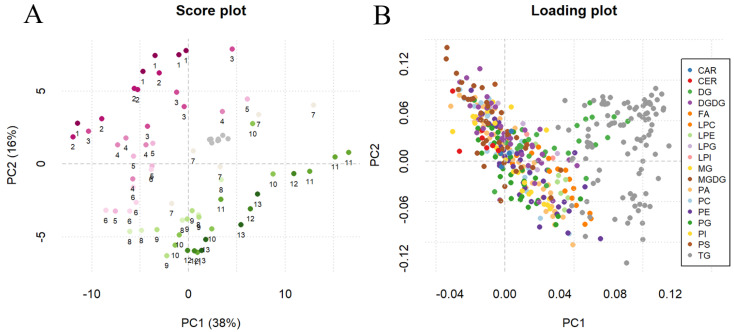
Score plot (**A**) and loading plot (**B**) of the Principal Component Analysis (PCA) on log-transformed and Pareto-scaled abundance of sampled grapes, collected at 13 different time-points from the onset of véraison throughout the ripening process. In the scores plot (**A**), in addition to the quality control (QC) samples (grey color), different colors represent a single time-point (numbers under each sample from 1 to 13), with inclusive biological replicates. In the loading plot (**B**) colors represent the different classes reported in the legend.

**Figure 3 metabolites-11-00827-f003:**
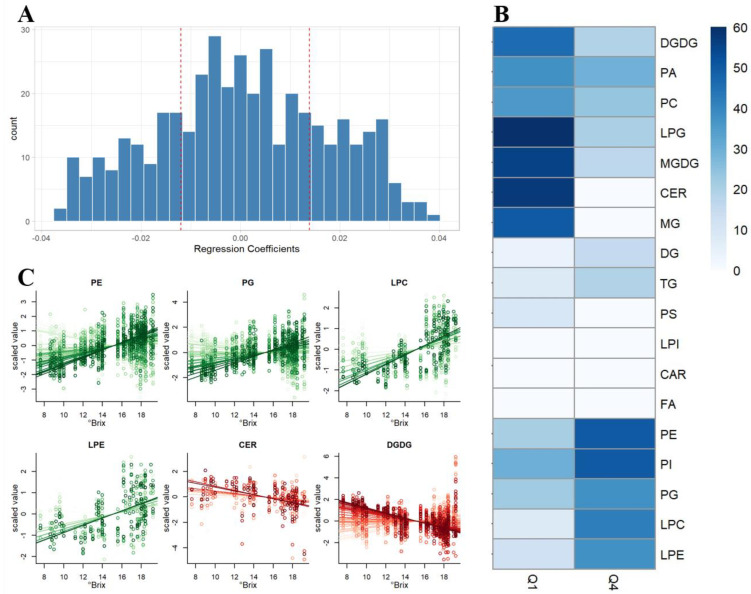
(**A**) Histogram of the regression coefficients distribution with Q1 and Q4 thresholds. (**B**) Heatmap of the quartile distribution of regression coefficients by lipid compound class. Proportions of compounds in first and fourth quartile are depicted with a color scale, where dark blue represents the categories with higher proportions, and light blue represents the lowest ones. (**C**) Behavior of some interesting class of compounds expressed as scaled values. Color is directly and inversely proportional to the corresponding regression coefficient for the classes with most of compounds located in the Q4 and Q1, respectively.

**Table 2 metabolites-11-00827-t002:** Method validation parameters.

Class	Compounds in Method	Based on the IS Compounds			Matrix	Validated	Based on the Reference Matrix		
		Recovery %	LOD (µg/g)	Linearity (µg/g)			Repeatability Range (CV%)	Intra-Day Range (CV%)	Inter-Day Range (CV%)
CAR	48	99	0.00003	0.0003–3	5	4	9–17	7–10	8–19
CER	210	118	0.005	0.03–150	11	7	8–15	2–15	6–21
DG	630	118	0.00003	0.0015–3	132	26	3–19	2–12	5–20
DGDG	630	96	0.00003	0.0003–30	43	37	2–18	2–15	6–21
FA	35	94	0.00003	0.003–300	8	5	9–19	4–13	6–18
LPA	35	76	0.05	0.15–300	0	0	--	--	--
LPC	35	100	0.00003	0.0003–3	12	12	4–9	2–7	5–7
LPE	35	98	0.00003	0.0003–150	8	8	2–7	2–4	4–7
LPG	35	29	0.00003	0.0003–150	5	5	5–19	3–8	4–11
LPI	35	4	0.00003	0.00015–300	5	2	11–14	14–15	17–19
LPS	35	39	0.003	0.015–300	0	0	--	--	--
MG	35	106	0.001	0.003–150	3	2	10–20	5–7	8–13
MGDG	630	100	0.00003	0.00015–3	150	36	2–17	1–14	4–21
PA	630	101	0.001	0.003–300	53	45	4–20	2–16	4–21
PC	630	105	0.005	0.015–300	51	25	3–20	3–16	10–21
PE	630	101	0.00003	0.0003–150	60	34	4–20	2–15	6–21
PG	630	92	0.0001	0.0003–30	104	32	4–19	2–12	5–21
PI	630	68	0.00003	0.0003–30	31	20	6–21	3–16	6–21
PS	630	103	0.0003	0.015–300	59	11	5–18	3–14	9–20
SM	35	81	0.0003	0.03–30	0	0	--	--	--
TG	1834	95	0.005	0.015–30	305	101	4–21	1–16	5–21
TOTAL	8077				1045	412			

Limit of detection, LOD; internal standard, IS; number of compounds identified in the reference matrix, #matrix; numbers of compounds successfully validated in the reference matrix, #validated.

## Data Availability

All the data that support the findings of this study are available within the manuscript, Appendix A, or from the corresponding authors upon request.

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
