# Peer review of "Grape Lipidomics: An Extensive Profiling thorough UHPLC-MS/MS Method"

_metabolites, 2021, doi:10.3390/metabo11120827_

Round 1
Reviewer 1 Report
The Authors present the profiling of lipids in grapes using UHPLC-MS/MS. Authors optimize the method that contains 8098 MRM. Such a huge amount requires repetition of analyses in positive and negative ionization mode. 755 compounds were identified and semi-quantified.
The work contains an interesting set of data and it may be published in Metabolites. However, some revision is necessary.
Points that need correction:
Have Author an idea about modification of the extraction procedure to extract sufficient LPG, LPI, and LPS? A different extraction method and focus on these three groups of compounds could complement the lipidome. It is rather the idea for the future, not to do it in this work.
XBridge Amide column (4.6 × 150 mm, 3.5 μm) is not a column for UHPLC, diameter to high. It is an HPLC.
The descriptions in Figure 1 are a too mental shortcut. I suggest writing down the full column names and naming the analytes correctly. This is probably about positive and negative ionization modes.
Author Response
Dear Reviewer,
Thank you very much for your valuable comments expressed regarding our submitted article. In the paragraphs below, please find addressed your remarks which were found to be extremely useful in improving the quality of our manuscript.
Q: Have Author an idea about modification of the extraction procedure to extract sufficient LPG, LPI, and LPS? A different extraction method and focus on these three groups of compounds could complement the lipidome. It is rather the idea for the future, not to do it in this work.
A: The preliminary work prior the method development included the use of various extraction techniques (data not presented in this manuscript), in addition to the Folch’s method. We are aware that this extraction method is influencing the effective recovery of lysoGPL species due to their high solubility in an aqueous phase. Several extraction procedures were reported in the literature [1], in order to achieve a successful recovery of lysoGPL species from the aqueous phase after extraction of biological samples with a modified Bligh–Dyer procedure for instance. But in the desire to extract as many lipids as possible with a single method, we opted for the aforementioned Folch's method with awareness of its shortcomings, which we finally mentioned in the manuscript. That said, thank you for your encouragement in our possible future work.
- Han, X. Lipidomics - Comprehensive Mass Spectrometry; Desiderio M., D., Ed.; Wiley, 2016;
Q: XBridge Amide column (4.6 × 150 mm, 3.5 μm) is not a column for UHPLC, diameter to high. It is an HPLC.
A: Yes, the XBridge Amide column is an HPLC column; We specified “HPLC“ in the section ‘3.3.1. Optimization of Liquid Chromatography Conditions’.
Q: The descriptions in Figure 1 are a too mental shortcut. I suggest writing down the full column names and naming the analytes correctly. This is probably about positive and negative ionization mode
A: We have tried to make the caption more accurate so as to help the interpretation of Figure 1. We have modified the legend as follows: “Figure 1: Distribution of the different compound classes along the retention time (RT) according to their m/z; the three tested columns (XBridge Amide column, Aquity CSH-C18 column and Aquity BEH-C18 column) are shown. Positive analytes: CAR carnitines; CER ceramides; DG diacylglycerols; DGDG digalactosyldiacylglycerols; LPC lyso-glycerophosphocholines; MG monoacylglycerols; MGDG monogalactosyldiacylglycerols; SM sphingomyelins; TG triacylglycerols. Negative analytes: FA free fatty acids;LPA lyso-glycerophosphates; LPE lyso-glycerophosphoethanolamines; LPI lyso-glycerophosphoinositols; LPG lyso-glycerophosphoglycerols; LPS lyso-glycerophosphoserines; PA glycerophosphates; PC glycerophosphocholines; PE glycerophosphoethanolamines; PI glycerophosphoinositols; PG glycerophosphoglycerols; PS glycerophosphoserines.”
We hope this will satisfy your requests of clarification.
With Kind Regards,
Giulia Chitarrini, for the authors
Reviewer 2 Report
The manuscript by Masuero et al. is a relatively straightforward report of their developed LC-MS/MS method applied to characterizing the grape lipidome. They provide a logical flow of testing and validation to establish what compounds to look for, how to separate, and how to quantify/identify. Following their setup, they apply this method to understanding more about the grape maturation process, in terms of lipid composition. The manuscript was very clean in my opinion, as the data is well put-together and analyzed. They presented their limitations well and did not overstate their conclusions. This is a useful reference dataset for both the grape lipidomics field as well as larger plant lipidomics field. A few minor suggestions are provided below for clarification.
- Check meaning in first use of reference 12, “hundreds of thousands”?
- Check meaning of statement around reference 24 as yeast doesn’t require to these lipids for general metabolism. They can use their Δ9-desaturase and that is sufficient for most strains; although I agree if the FA are bioavailable/fed they can be utilized.
- A few of the supplemental tables could benefit from enhanced legend to make them standalone.
- The use of semi-quantitative and MS threw me off given the report µg/g of fresh grape powder. I guess it is a matter of personal preference, but I’d still consider this a quantitative method, just with limitations on the exact/true number.
Author Response

(The authors gave the same response as above.)

Reviewer 3 Report
In this article, a UPLC-MS method was developed to determine the lipid profile of grape samples. A triple quad instrument was applied, the MS method involves numerous MRM analyses. The developed method proved to be efficient for the analysis of most lipid molecules, showing acceptable recovery for most of the components. As an application for the developed method grape samples were analyzed. The experiments and the article are well organized and well designed. The overall merit of the article is good.
The recoveries are in a wide range. Please compare these recovery values with other results. Was there any further correction of concentrations before data processing? If not why it is not relevant?
I suggest creating a supplementary document. Without that, it is hard to navigate in the uploaded supporting data.
Author Response
Dear Reviewer,
Thank you very much for your comments and questions. Please find our answers to them in the paragraphs below.
Q: The recoveries are in a wide range. Please compare these recovery values with other results.
A: The recoveries obtained appeared to be good for almost all the classes of compounds, with the exception for LPG, LPI, and LPS. Since the used Folch’s extraction method is influencing the effective recovery of aforementioned lysoGPL species due to their high solubility in an aqueous phase. We could achieve a successful recovery of lysoGPL species from the aqueous phase after extraction of biological samples with a modified Bligh–Dyer procedure for instance. Thus, a compromise was needed for the use of only one extraction method, which, however, proved to be satisfactory, with the mentioned disadvantages, which we stated in the manuscript “We are aware that the extraction method used was a sort of compromise and cannot be optimized for all the compound classes together, but this situation has to be accepted to ensure a large coverage of the different lipid classes.”
Q: Was there any further correction of concentrations before data processing? If not, why it is not relevant?
A: In section ‘3.6. Data Analysis’ we added the following sentence to specify how data have been calculated: “The compound semi-quantification has been calculated using one internal standard for each class (Table 1) and normalized for the exact weight of grape powder.”
Q: I suggest creating a supplementary document. Without that, it is hard to navigate in the uploaded supporting data.
A: Thank you for your suggestion. To help the readers we created a single Excel file which includes 4 supplementary tables.
Best regards and thank you again.
Giulia Chitarrini, for the authors
Reviewer 4 Report
The manuscript by Masuero, Skrab et al. describes the development of a UHPLC-MS/MS lipidomics method for semi-quantification of hundreds of lipid compounds in grapes. This assay was successively used to investigate the variation of the lipids profile during the maturation process and how this profile could be associated (predict) to the °Brix rate.
The manuscript is very well written, the methods well described and detailed as well as the results. Furthermore, the developed lipidomics assay might be applied when having other kinds of samples.
I have very few minor comments to address:
1) Method section
- In the method section is not address how QC samples were prepared (I guess it was a pool of all the samples?)
- It is not mentioned how the Brix measurement was done and it is not reported in any table.
- LOD determination. I would say that there are different ways of calculating it. Writing that it was based on a visible peak, it is a bit subjective. The standard rule would be 3.3x(S/b) and for the LOQ 10x. I would consider recalculating it based on the guidelines.
2) Results
- Figure 2. It is not easy to follow without a legend for the different colors (better would also be with one color for each time point instead of a gradient scale).
- Figure 3. I would remove the panel A, it does not give a lot of info.
Author Response
Dear Reviewer,
Thank you for your very nice comments regarding our manuscript. We addressed your comments in the following paragraphs.
Q: In the method section is not address how QC samples were prepared (I guess it was a pool of all the samples?).
A: Yes, the QC is a pool of the extracted samples, thank you for this observation. We added the following sentence in the material and methods section: “A quality control (QC) sample has been created by pooling the samples using 20 µL of each extract, used as QC and injected in the same sequence as the samples.”
Q: It is not mentioned how the Brix measurement was done and it is not reported in any table.
A: Thank you for your observation. Regarding the °Brix measurement, we inserted the following phrase in the Materials and Methods, section 3.4. Sample Collection and Lipid Extraction: “Total soluble solids (°Brix) were measured using a manual refractometer (ATC-1, Atago, Tokyo, Japan).” The °Brix, however, are reported on top of the Table S3, together with sample name, sample date and number of replicas.
Q: LOD determination. I would say that there are different ways of calculating it. Writing that it was based on a visible peak, it is a bit subjective. The standard rule would be 3.3x(S/b) and for the LOQ 10x. I would consider recalculating it based on the guidelines.
A: We changed Table 2 recalculating the new LOD as suggested following the 3.3x(S/N). For the quantification we decided to keep the linearity range as limits.
Q: Figure 2. It is not easy to follow without a legend for the different colors (better would also be with one color for each time point instead of a gradient scale).
A: We tried to build the figure with one color for time points but the trend over the time was not visible; for this reason, we decided to divide the points in two groups (first maturation part from point 1 to 7; second maturation part from point 8 to 13) to emphasize the visible trend.
Q: Figure 3. I would remove the panel A; it does not give a lot of info.
A: We would like to keep Figure 3A in order to show to the reader the distribution and magnitude of regression coefficients for all compounds. With that figure we think that we are able to provide information about the proportion of compounds with the highest or lowest regression coefficients (as well as, that there were compounds correlated both directly and inversely with the °Brix degree), and also where would be the cutoff of the percentiles 25 and 75.
I hope this will satisfy your requests of clarification.
With Kind regards,
Giulia Chitarrini, for the authors
Reviewer 5 Report
The authors used lipidomic analysis to describe the composition of the gape lipidome using HPLC-MS/MS. The quantification has been validated and is sufficiently accurate for the comparing of variously ripe grapes. I think, problems are in identification of lipids. The data were well statistically processed, and the result was discussed.
The authors write that they focused on fatty acids with chemical backbones made up of an even number of carbons, from 14 to 22. I also consider that FA from 14 to 22 will mostly contains in lipids. Subsequently, based on the results of Figure S1, they discuss, … “The obtained result could therefore indicate that grapes are dominated by lipids with long-chain (aliphatic tails of 14 to 22 carbons) fatty acids,…..” In my opinion, this cannot be said, when they were not measuring anything else. Please correct it.
The complete identification of many lipid classes based only on a single MRM transition is unreliable. Without knowledge of the exact mass or more fragments (at least characteristic ion of the polar group) they may be at fault identification. In addition, when fragmenting glycerophospholipids in the negative mode, both FA are cleaved (from sn-1 and sn-2 positions). The sn position cannot be determined, and an underscore should be given between the acids. Example: PC 18:0_18:1 is correct abbreviation, when sn positions are not identified.
Similarly problem is for for DGs and TGs (sn-3 position) in positive ion mode.
The sn positions can be determined from the ratios of these FAs related fragments.
Some polyunsaturated FA identified in lipids (for example 14:6, 14:5, 16:6,…) are very unusual and identification based on one MRM without any other information (retention order in lipid class, MS2 product spectra) is very weak. In my point of view, these identifications should be supported by some independent method. For example, by profiling and identification of the fatty acid after transesterification of the lipids.
Small comments:
It is recommended to use two-letter abbreviations (TAG … TG, DAG … DG, etc.)
The abbreviations TAG in Table S3 are not common. It is necessary to at least explain them. What does it mean? FA at the end of the abbreviation describe identified FA.
Figure 2A - I miss legend. Sampling over time is not clear.
Table S2. - Description of the numbers missing… % of VC?
Author Response
Dear Reviewer,
Thank you very much for your comments and constructive observations made regarding our manuscript. In the paragraphs below, we have addressed your remarks in hope of clarifying all the questions, regarding submitted manuscript.
Q: The authors write that they focused on fatty acids with chemical backbones made up of an even number of carbons, from 14 to 22. I also consider that FA from 14 to 22 will mostly contains in lipids. Subsequently, based on the results of Figure S1, they discuss, … “The obtained result could therefore indicate that grapes are dominated by lipids with long-chain (aliphatic tails of 14 to 22 carbons) fatty acids...” In my opinion, this cannot be said, when they were not measuring anything else. Please correct it.
A: We converted the questionable phrase into: “As it has been previously reported in the literature [34], the plant tissues most often contain between 14 and 24 carbon atoms, which could confirm our decision regarding the chain length for the validation.”
Q: The complete identification of many lipid classes based only on a single MRM transition is unreliable.
A: The method has an extensive number of MRMs. To guarantee a suitable number of points per peak along the chromatographic run it was not possible to add more than one MRM per compound. Moreover, even with one MRM per compound, separate positive and negative methods have been built.
Q: Without knowledge of the exact mass or more fragments (at least characteristic ion of the polar group) they may be at fault identification.
A: We thank the reviewer for this comment. We were not able to build a method using more than one MRM per compound due to the instrument limitations. For that reason, we decided to increase the compounds annotation confidence. To evaluate and validate the compound annotation, we decided to use the KMD values as reported in recent lipidomics literature [1]. As reported by Lipid Maps website https://www.lipidmaps.org/tools/ms/kendrick_form.php, “in high-resolution mass data KMD may conveniently be used for identification of members of a homologous series of various classes of lipids. The Kendrick mass is defined by setting the mass of a chosen molecular fragment, typically CH2, to an integer value in atomic mass units:
The Kendrick mass defect (KMD) is defined as the difference between the exact Kendrick mass and the nominal Kendrick mass (integer):
In our paper we used the retention time plotted versus the KMD value as reported in recent papers. In the track changes version, we added the explanation in the material and method section, and we explained the followed criteria for the validation. We also added a Supplementary Figures to show the 412 selected compounds which follow the criteria for the annotation confidence. We then reperformed the data analysis of the application considering only those 412 compounds.
- Lange, M.; Angelidou, G.; Ni, Z.; Criscuolo, A.; Schiller, J.; Blüher, M.; Fedorova, M. AdipoAtlas: A reference lipidome for human white adipose tissue. Cell Reports Med. 2021, 2, 100407, doi:10.1016/J.XCRM.2021.100407.
Q: In addition, when fragmenting glycerophospholipids in the negative mode, both FA are cleaved (from sn-1 and sn-2 positions). The sn position cannot be determined, and an underscore should be given between the acids. Example: PC 18:0_18:1 is correct abbreviation, when sn positions are not identified.
Similarly, problem is for DGs and TGs (sn-3 position) in positive ion mode.
The sn positions can be determined from the ratios of these FAs related fragments.
A: Thank you for your comment, we changed the compound abbreviation by adding an underscore between the acids as suggested.
Q: Some polyunsaturated FA identified in lipids (for example 14:6, 14:5, 16: 6…) are very unusual and identification based on one MRM without any other information (retention order in lipid class, MS2 product spectra) is very weak. In my point of view, these identifications should be supported by some independent method. For example, by profiling and identification of the fatty acid after transesterification of the lipids.
A: As suggested we decided to apply the Kendrick mass defect (KMD) to confirm the compound annotation. We plotted the KMD value of each compound versus the retention time; the compounds out of the diagonal (different carbon number with the same double bounds) and horizontal (same carbon number but different double bounds) have been excluded; among the excluded there were some unusual compounds that have not been included in the application.
Q: It is recommended to use two-letter abbreviations (TAG … TG, DAG … DG, etc.).
A: We followed your suggestion changing the abbreviation: TAG--TG; DAG--DG; MAG--MG; FFA—FA as reported in track change version.
Q: The abbreviations TAG in Table S3 are not common. It is necessary to at least explain them. What does it mean? FA at the end of the abbreviation describe identified FA.
A: We explained the TAG (TG in the new draft version) abbreviation in the footnotes of the Table S3 as follow: “Triacylglycerol (TG) annotation is given by the sum of the carbon followed by the specific fatty acid (FA) lost identified by the MRM transition.”
Q: Figure 2A - I miss legend. Sampling over time is not clear.
A: We tried to build the figure with one color for time points but the trend over the time was not visible; for this reason, we decided to divide the points in two groups (first maturation part from point 1 to 7; second maturation part from point 8 to 13) to emphasize the visible trend. We specified this into the main text in the track change version and we specified the maturation point numbers in the capture.
Q: Table S2. - Description of the numbers missing… % of VC?
A: We have formatted the Table S2 to make the description more visible.
We hope that our explanations will satisfy your requests of clarification.
Kind regards,
Giulia Chitarrini, for the authors
Round 2
Reviewer 5 Report
In my opinion, the manuscript has been improved significantly. I am satisfied with the answers. The level of certainty of identification has been increased to an acceptable level.